# Occurrence of Types A and B Trichothecenes in Cereal Products Sold in Romanian Markets

**DOI:** 10.3390/toxins15070466

**Published:** 2023-07-20

**Authors:** Adrian Maximilian Macri, Andras-Laszlo Nagy, Sorana Daina, Diana Toma, Ioana Delia Pop, George Cosmin Nadăș, Adriana Florinela Cătoi

**Affiliations:** 1Department of Animal Nutrition, Faculty of Veterinary Medicine, University of Agricultural Sciences and Veterinary Medicine of Cluj-Napoca, 400372 Cluj-Napoca, Romania; adrian.macri@usamvcluj.ro; 2Department of Biomedical Sciences, Ross University School of Veterinary Medicine, Basseterre P.O. Box 334, Saint Kitts and Nevis; anagy@rossvet.edu.kn; 3Department of Veterinary Toxicology, Faculty of Veterinary Medicine, University of Agricultural Sciences and Veterinary Medicine of Cluj-Napoca, 400372 Cluj-Napoca, Romania; 4Department of Biochemistry, Faculty of Veterinary Medicine, University of Agricultural Sciences and Veterinary Medicine of Cluj-Napoca, 400372 Cluj-Napoca, Romania; 5Department of Land Measurements and Exact Sciences, Faculty of Horticulture, University of Agricultural Sciences and Veterinary Medicine of Cluj-Napoca, 400372 Cluj-Napoca, Romania; popioana@usamvcluj.ro; 6Department of Microbiology, Immunology and Epidemiology, Faculty of Veterinary Medicine, University of Agricultural Sciences and Veterinary Medicine of Cluj-Napoca, 400372 Cluj-Napoca, Romania; gnadas@usamvcluj.ro; 7Department of Pathophysiology, Iuliu Hațieganu University of Medicine and Pharmacy, 400012 Cluj-Napoca, Romania; adriana.catoi@umfcluj.ro

**Keywords:** trichothecenes, bread, bakery products, wheat, pasta, GC-MS, Romania

## Abstract

In view of the frequent occurrences of mycotoxins in cereals, this study assessed the presence of trichothecenes in 121 samples from Romanian markets. These samples were divided into five groups based on product type: (1) bread and bakery products containing white flour, (2) half-brown bread with whole wheat flour, (3) brown bread containing rye flour, (4) pasta, and (5) raw wheat. Gas Chromatography-Mass Spectrometry was used to detect 13 different mycotoxins, which included the Type A compounds HT-2 toxin and T-2 toxin, as well as the Type B compounds deoxynivalenol and nivalenol. Results indicated trichothecene contamination in 90.08% of our samples, with deoxynivalenol predominating by at least 78% in each examined group. Co-occurrence of three or four trichothecenes were found in 23.85% of our samples. Our study underscores the necessity of consistent monitoring of staple foods to prevent the intake of harmful trichothecenes by consumers.

## 1. Introduction

For decades, mycotoxin contamination has been an important area of research. Mycotoxins in agricultural commodities have long been considered a risk to both human and animal health [1,2]. Due to cultural and economic considerations, cereal and products derived from cereals are considered very important types of food staples [3]. Wheat is one of the world’s most significant cereal crops, providing carbohydrates, proteins, minerals, lipids, vitamins, and fiber [4]. Europe is the largest wheat and wheat products consumer, with Romania being the fourth major producer of wheat in Europe [1,5].

The occurrence of mycotoxins in worldwide agricultural commodities has been widely recorded. This is the consequence of the natural contamination of grains by a variety of fungal pathogens that are responsible for serious diseases. Therefore, consumers are potentially exposed via food to a large number of mycotoxins [6]. *Fusarium* species are widely recognized as important wheat pathogens. Beyond pathogenicity, species including *Fusarium sporotrichioides*, *F. graminearum* and *F. equiseti* are capable of synthesizing a toxic class of tetracyclic sesquiterpenoid compounds called trichothecenes [7,8]. Trichothecenes comprise 150 mycotoxins, each with characteristic rings that are divided into two groups: macrocyclic and non-macrocyclic. Non-macrocyclic rings are common in food matrices and are further categorized as type A or type B [1,9,10,11].

The *Fusarium* toxins most commonly reported in food from group A trichothecenes are HT-2 toxin (HT-2), T-2 toxin (T-2), diacetoxyscirpenol (DAS) and neosolaniol (NEO), while those from group B trichothecenes are deoxynivalenol (DON), nivalenol (NIV), 3-acetyldeoxynivalenol (3-DON), 15-acetyldeoxynivalenol (15-DON), fusarenon-X (FUS-X) and deoxynivalenol-3-glucoside (D3G) the most common masked mycotoxin [12,13,14,15,16,17,18,19,20]. The growth of *Fusarium* and the production of mycotoxins, including T-2 and HT-2 toxins, depends on multiple factors, the most important being temperature and humidity [21]. 

The acute symptoms of trichothecene damage by ingestion in humans are gastroenteritis, anorexia, nausea, and growth retardation. The effect on animal health depends on a number of factors, including the level and duration of exposure, the toxicity of the ingested compound, species, body weight, and age. Trichothecenes can modify their initial structure and can be found in their derived forms, which are not detectable in standard analyses. These redefined structures could be the result of food processing or plant metabolism [22].

According to the most recent European legislation, whether the mycotoxins are identified in raw material or processed food, numerous maximum limits have been established, depending on the type of mycotoxin and the foodstuff in question [23]. The maximum allowable limits for DON are currently in place for unprocessed cereals, durum wheat, oats, unprocessed maize, flour and pasta, bread, pastries, cookies, cereal snacks, and breakfast cereals [24]. Although various studies have shown that T-2 and HT-2 toxins are dangerous to humans, no official limit values have yet been established within the European Union. According to the Joint FAO/WHO Expert Committee on Food Additives (2001), the provisional maximum tolerable daily intake for a single mycotoxin or the total of T-2 and HT-2 is 60 ng/kg body weight per day [25]. An acceptable daily intake of 100 ng/kg body weight was established in Europe by the European Food Safety Authority [26]. However, these different types of acceptable mycotoxin levels established for both raw materials and foodstuffs are reflected in the change in mycotoxin levels during food manufacturing.

Analytical methods were developed over time for mycotoxins monitoring of food and feed commodities to ensure that levels proposed by government regulations are met. Scientific literature proposes several available methods, from immunoassays, including enzyme-linked immunosorbent assays (ELISA) and fluorescence polarization immunoassays [27] to metabolomics approaches such as liquid chromatography coupled with tandem mass spectrometry (LC-MS), high-performance liquid chromatography coupled with ultraviolet (HPLC-UV), fluorescence detector (HPLC-FL), and gas chromatography coupled with mass spectrometry (GC-MS). Co-occurrence in the same food of various trichothecenes requires simultaneous determination methods which allow a sensitive, accurate and real-time analysis. Before analysis, due to the complexity of food and feed matrices, extraction procedures are necessary and currently, there are several developed techniques such as solid–liquid extraction (SLE) with appropriate solvents, QuEChERS (Quick Easy Cheap Effective Rugged Safe) [28,29,30,31,32].

The presence of different types of trichothecenes has been reported over the years in various food samples from different countries. Most frequently, the contamination of maize with trichothecenes was reported [33]. Although wheat contamination with trichothecenes is frequently occurring, to the best of our knowledge, only a few studies are available for Romania with limited focus on a small number of mycotoxins or based on enzyme-linked immunosorbent assay [34]. Tabuc et al. conducted a study on maize, barley, and wheat samples and determined the presence of DON, aflatoxin B_1_, zearalenone, fumonisins, and ochratoxin A using ELISA assay [35]. In 2017, another study was performed for DON determination in Romanian cereal crops with an ELISA assay [36].

Monitoring exposure to various mycotoxins has become a key part in order to ensure food safety. The main objective of the present study was to evaluate the de occurrence of trichothecenes mycotoxins from several staple foods consumed in Romania, such as bakery products, pasta, and wheat. Therefore, the GC-MS method was used in order to simultaneously determine the presence of 13 trichothecenes DON, NIV, SCIRP, T2 TETRAOL, FUS-X, MAS, 15-ADON, 3-ADON, T2 TRIOL, NEO, DAS, HT-2, T-2 in white bread, half-brown bread and brown bread (to cover all consumers’ preferences), pasta, and wheat samples commercialized in Romania.

## 2. Results

### 2.1. Detection Limits, Recoveries

Investigations were carried out in order to assess the incidence of trichothecenes mycotoxins in bakery products, pasta, and wheat, considered as staple foods in Romania. For this purpose, the GC-MS method was tested for recoveries and repeatability in wheat, bread, and pasta matrices, for all 13 compounds. The results are summarized in Table 1.

Recovery rates varied, across food matrices that were used: wheat, bread, and pasta. Generally, good recoveries were obtained in bread, comparable recovery values were observed in pasta and wheat for both spiking concentrations.

### 2.2. Occurrence of Trichothecenes in Analyzed Samples

The GC-MS method was applied to determine de presence of thirteen type A and type B trichothecenes in wheat, bread, and bakery products and pasta commercialized in Romania. A summary of the contamination in analyzed samples is presented below (Table 2):

#### 2.2.1. DON Occurrence

Table 3 shows the median values and percentiles 25 and 75 of DON. The highest level was found in group B {192 µg/kg (78–268)} and the lowest value in group D {21 (15–30)}. In samples from group A (white bread), contamination with DON registered levels from 15 µg/kg to 352 µg/kg, and in group B (half brown bread) from 50 µg/kg to 346 µg/kg. 

In order to identify whether there are significant differences between A-D groups, Kruskal–Wallis test was applied. Significant differences were identified between the groups (*p* < 0.001) (Figure 1).

The differences’ level of significance between each two groups within the A-D groups were analyzed by using Pairwise Comparisons. Table 4 and Figure 2 show the significant/non-significant differences between the groups. The level of significance has been adjusted by the Bonferroni correction for multiple tests.

Significant statistical differences were observed between group D (pasta) and group A (white bread), and between group D and group B (half brown bread) and C (brown bread), respectively. 

When data from group A was compared with data from group C and from group B, respectively, and data from group C was compared with data from group B, non-significant differences were observed.

Applying the Shapiro-Wilk test for DON in wheat samples, it was observed that the values were not normally distributed. Table 5 shows the descriptive statistics regarding DON occurrence in wheat samples from different counties of Romania. The median values and percentiles 25 and 75 of DON were determined. The highest level was found in subgroup E_3_ (Cluj county) {509.50 µg/kg (169.25–933.75)}. The lowest level was found in subgroup E_1_ (Timis county) {205.50 µg/kg (100.75–387)}. Subgroup E_2_ registered a value of 502.50 µg/kg (220.75–1300.50).

Kruskal–Wallis test was applied in order to compare data from all three groups of wheat samples coming from Timis, Alba, and Cluj county regarding DON occurrence. No significant differences between the groups were identified (*p* = 0.145) (Figure 3).

#### 2.2.2. 15-ADON

As regards the occurrence of 15-ADON in analyzed samples, groups A, B, C, and D were free of 15-ADON. In wheat samples coming from Timis county, only two samples were contaminated, and the values were 6 µg/kg and 9 µg/kg, respectively. Therefore, in this case, no statistical analysis was applied. 

Table 6 presents the descriptive statistics for 15-ADON occurrence in wheat samples. The median value corresponding to Alba county was 9 µg/kg (6–57).

The highest concentration of 15-ADON registered in samples coming from Cluj county had a value of 52 µg/kg. 

We observed no significant differences with regard to 15-ADON median values between wheat samples from Timis, Alba, and Cluj counties (Kruskal–Wallis test, *p* = 0.112) (Figure 4).

#### 2.2.3. HT-2 Occurrence

From all analyzed samples, HT-2 occurred in 21 wheat samples, from all three counties, and in one sample (bio bread) from group A where the concentration was 3 µg/kg. None of the samples from groups B (half brown bread), C (brown bread), and D (pasta) were contaminated with HT-2. 

Table 7 shows the normally distributed data for each county with mean ± standard deviation. In Timis county samples, the mean concentration of HT-2 was 5 µg/kg the mean concentration of HT-2 was of 6.25 µg/kg in Alba county samples and 9.2 µg/kg in samples coming from Cluj county, respectively. 

Significant differences were noted between Timis, Alba, and Cluj counties with regard to HT-2 values (One-Way ANOVA test *p* = 0.043, F = 3.782). 

The Post Hoc analysis by using Tukey HSD (honestly significant difference) test (Table 8) showed significant differences between wheat samples coming from Timis county as compared to those coming from Cluj county. For the other comparisons between groups, no significant differences were observed.

#### 2.2.4. Contamination with Other Trichothecenes

From the total of 121 analyzed samples, 90.08% (109) were contaminated with one (77.06%), two (11%), three (10.09%), or four trichothecenes (1.83%). From all samples subjected to our study, 12 (9.92%) were free of contamination with trichothecenes, as follows: three samples from group A (1 croissant, 1 white baguette, and 1 bread), one sample from group B (graham bread), four samples from group D (1 tortiglioni pasta, 1 spirali pasta, 1 conchiglie pasta, and 1 spaghetti pasta) and four samples from group E (one sample of wheat from Timis county and three wheat samples from Alba county).

From all 13 trichothecenes investigated in the analyzed samples, SCIRP, T2-TETRAOL, FUS-X, MAS, 3-ADON, T2-TRIOL, and NEO were not detected in any group of samples. DAS was found only in one wheat sample coming from Alba county (subgroup E2) with a concentration of 19 µg/kg. NIV occurred only in one sample of wheat originating from Cluj county (subgroup E3), in a concentration of 30 µg/kg. As regards T-2 mycotoxin, this was identified in two samples, one bio bread (group A) with a concentration of 5 µg/kg, and one wheat sample from Alba county (subgroup E2) with a concentration of 7 µg/kg. The same bio bread sample where T-2 mycotoxin was detected, was contaminated with both DON and HT-2, in concentrations of 48 µg/kg and 3 µg/kg, respectively. 

## 3. Discussion 

Mycotoxin analysis performed by the GC-MS method permitted a multi-mycotoxin determination from a variety of samples. Detection and quantification limits were determined using purified wheat extract and values ranged from 2 µg/kg to 45 µg/kg for LOD and from 6 µg/kg to 137 µg/kg for LOQ. These values were in good agreement with LOD and LOQ limits mentioned by other authors: values between 2–12 µg/kg for LOD and between 6–38 µg/kg for LOQ [37], from 7 µg/kg to 65 µg/kg (LOD) and from 20 µg/kg to 196 µg/kg for LOQ [38] and values of 16 µg/kg for LOD and 50 µg/kg for LOQ [39]. The recovery rates that we obtained in our experiment by using spiking concentrations of 500 µg/kg and 200 µg/kg were between 60–110% for DON and from 60% to 130% for T-2 and HT-2, in accordance with the specific requirements proposed by the European Commission regulation [40], and also in good agreement with recoveries reported previously in other studies. When recoveries were tested at 20, 100, and 1000 µg/kg levels for several trichothecenes in barley samples, values ranged from 56% to 105%, with the lowest recovery values registered for NIV (56.3% to 72.4%). A decrease in recovery rate was observed due to the increasing spiking concentration [28], as we found in our study for the recovery rates of NIV, with values of 55% in wheat, 68% in bread and 57% in pasta at spiking concentration of 500 µg/kg. Generally, recoveries obtained at a spiking level of 200 µg/kg showed slightly improved values. Another study showed that at a spiking level of 500 µg/kg in wheat, recoveries ranged from 73% to 91%, the lowest recovery rates being obtained for 15-ADON (73%) and for NIV (74%) [37]. Rodriguez-Carrasco et al. obtained recovery rates from 74% to 124% for NIV, FUS-X, DAS, 3-ADON, NEO, DON, T-2, and HT-2 at spiking concentrations of 20, 40, and 80 µg/kg in wheat semolina [41]. Performing spiking at a concentration of 100 µg/kg for DON, 15-ADON, and NIV in wheat flour, the authors obtained recoveries of 103%, 93%, and 61% [42]. The low recovery rate in the case of NIV could be explained by the very volatile character of this molecule, compared with the other mycotoxins [28]. Taking into account the results that we obtained for the method validation regarding recoveries and limits of detection and quantification, we consider that this analysis is sensitive and precise.

After investigating our samples, we observed that, among all the studied trichothecenes, DON represented the predominant mycotoxin in all analyzed groups and the total contaminated samples (n = 109) contained DON in various concentrations. From the total of 32 samples pertaining to group A (white bread and bakery products containing only white wheat flour), 90.62% (n = 29) were contaminated. From the total of 14 samples corresponding to group B (half brown bread which contains whole wheat flour), 92.85% (n = 13) were contaminated, and samples from group C (brown bread which contains rye flour) were all contaminated (100%). This was in contrast with a study performed by Schollenberger et al., where a lower incidence of DON in bread containing rye flour compared to bread with wheat flour was observed [43]. From group D (pasta), 78.94% (n = 15) of samples presented contamination. As concerns wheat samples (group E), from the total of 42 samples, 90.47% (n = 38) were contaminated with DON. The co-occurrence of DON, 15-ADON, and HT-2 was observed in several wheat samples, although the co-occurrence of 3-ADON was previously observed [44]. Moreover, 65.14% (n = 71) of the samples contaminated with DON were represented by bread and bakery products, respectively pasta. Results displayed a degree of contamination from 15 µg/kg (rolls) to 352 µg/kg (white round bread), with a mean value of 96.65 µg/kg corresponding to samples from group A; from 50 µg/kg (graham bread) to 346 µg/kg (whole wheat toast bread) with a mean value of 178.38 µg/kg occurring in samples from group B; from 15 µg/kg (brown baguette) to 346 µg/kg (wheat-rye bread) observed in group C; samples from group D varied from 15 µg/kg (cornetti rigati pasta, conchiglie pasta, and macaroni) to 35 µg/kg (spaghetti). The contents obtained here suppressed neither the maximum accepted level of 500 µg/kg for bread and bakery products nor the limit of 750 µg/kg for pasta imposed by European regulation in force [24]. With regard to DON contamination data corresponding to bakery products analyzed from group A, only 24.13% registered values above 100 µg/kg, as compared with concentrations observed in samples from group B (bakery products from whole wheat flour), where 69.23% of the positive samples registered values above 100 µg/kg. This can be explained by the fact that DON concentration is higher in bran [45]. A study regarding DON presence in cereal-based food products from the Czech market reported a concentration range of 13–350 µg/kg in white flour products [46]. Another published study documented that DON content in pasta was higher (137.1 µg/kg) in comparison with DON content in bread, while for whole wheat bread contamination was up to 40.7 µg/kg, [2,47,48] findings that are in contrast with our experimental data. DON was also detected in baby foods based on cereals, in significant amounts of 29–270 µg/kg even exceeding the admitted level of 200 μg/kg for this type of food [32]. 

Some authors reported a contamination level of 3390 µg/kg in wheat grain from Timis county were intended for animal feed [49]. The results we obtained by analyzing wheat samples were comparable with those reported by other studies: in 92 samples of wheat from Poland DON levels were within the range of 10.5–1265.4 µg/kg [50]. Xu et al. performed a study on 370 wheat grains from China and observed an incidence level of 100%, with values between 109 µg/kg and even up to 86,255.1 µg/kg [51]. 

Other researchers did not detect 15-ADON in analyzed samples [29]. However elsewhere various concentrations from 0.62 µg/kg to 6 µg/kg were reported [52], and between 6 µg/kg to 30.6 µg/kg [22]. The co-occurrence of mycotoxins in the same sample could create a synergism regarding their toxicological effects. This represents a major public health concern regarding food safety and that requires more attention in assessing the interactions among mycotoxins present in food with further impact on human health [22].

NIV had a very low incidence, occurring only in one of the analyzed samples (one sample of wheat from Cluj county) in a concentration of 30 µg/kg, similar to other authors who reported NIV in breakfast cereals (31 µg/kg) [46], in cereal grain with an average content of 35 µg/kg [53] or in durum wheat (mean 32.32 µg/kg) and bread wheat (mean 5.12 µg/kg) [6]. T-2 toxin content was observed by González-Osnaya et al. in bread, with a concentration of 68.37 µg/kg [2], much higher than the values observed in our study. HT-2 values ranged between 9.06–34.43 µg/kg in ready-to-eat foods [29] while in our study HT-2 was predominant in wheat samples. As concerns DAS occurrence, some authors reported levels up to 176.3 µg/kg in wheat, while SCIRP, 3-ADON, MAS, and FUS-X were detected neither in wheat, oats or cereal products intended for children [53].

To prevent human exposure, authorities perform quarterly inspections of a maximum of three units (supermarkets and specialty stores, milling units and so on) in each county and whenever the situation requires the assessment of the presence of DON in bread and bakery products, pasta, and wheat and also for other mycotoxins is conducted. The National Sanitary Veterinary and Food Safety Authority (ANSVSA) has implemented the regulation where it specifies the methodological rules for the surveillance and control program for the safety of non-animal origin food in conformity with the European and national legislation for contaminant content [54]. As Romania is a representative country regarding cereal production and as cereal products are either exported or intended for the Romanian population, the need of ensuring safe food is continuous since some of the analyzed samples exceeded the maximum permitted level. 

## 4. Conclusions

The Evaluation of thirteen trichothecenes from wheat, bread, and bakery products as well as pasta was performed by using the gas chromatographic method. From the total of 121 samples subjected to analysis, 90.08% were contaminated with trichothecenes. DON represented the predominant mycotoxin, occurring in 90.62% of bakery products which contain white flour, in 92.85% of bakery products which contain whole wheat flour, 100% of bakery products which contain rye flour, as well as in 78.94% of pasta samples, and in 90.47% of wheat samples. Although samples represented by bakery products containing whole wheat flour were contaminated in higher concentrations than samples from white flour, none of them displayed exceeded the maximum level imposed by regulation. As regards wheat samples, only two of them surpassed the maximum admitted level for DON. HT-2 toxin showed an incidence of 20.18%, while 15-ADON presented an incidence of 13.76%. In 65.78% of the positive samples, we observed that two, three, or four mycotoxins co-occurred. Evaluation of multi-mycotoxins in different food matrices offers a reliable tool for risk assessment since their co-occurrence is possible and enhances the exposure to human health risks. These results point out the necessity of consistent control in order to prevent mycotoxin intake by the Romanian population through cereal-based foods. 

## 5. Materials and Methods

### 5.1. Samples 

A total of 121 samples of wheat, bakery products, and pasta were collected randomly from the Romanian market. Sample collecting represented approximately the proportion of various sorts of bread available on the Romanian market, as well pasta and wheat. All collected samples were divided into 5 groups, listed according to their ingredients:

Group A: white bread, n = 32. This group comprises bakery products obtained from white wheat flour (as main ingredient) types 650, 550, and/or 000. This group is represented by baguette, rolls, ciabatta, pretzel, croissant, bread with potato, bio bread, and white round bread. Beside wheat flour, other additional ingredients are possible: potatoes, yeast, water, salt, and food preservatives.

Group B: half-brown bread, n = 14. This group represents any bread/bakery products which contains whole wheat flour, graham, including dietary bread, graham bread, and whole wheat dietary bread. The main ingredient is represented by wheat flour type 650, with graham flour in different percentages, more often 21% of the total flour content. Other possible ingredients are salt, yeast, water, sunflower oil, and vinegar. 

Group C: brown bread, n = 14. In this category were included bread and bakery products formulated with different amounts of rye flour. 

Group D: pasta, n = 19. Here were included various types of pasta formulated from wheat flour and other additional ingredients such as egg, salt, and water. 

Group E: wheat, n = 42 from which three subgroups were divided, according to the county origin: E1, n = 13 (Timis county), E2, n = 11 (Alba county), E3, n = 18 (Cluj county).

Of the total number of samples, 42 are the wheat samples obtained from mills in three counties: Alba, Cluj, and Timis. The samples were collected with sampling spears and scoops. Three incremental samples were drawn from each top, middle, and bottom layers in the crop container (five sampling sites for each layer). Multiple subsamples (~500 g) were pooled, thoroughly mixed, and quartered. Opposite quarters were rejected, and the remainder was remixed thoroughly to obtain a final reduced sample of 250 g.

Bread and pasta were purchased from various supermarkets and bakeries. The bread samples were dried at 40 °C before milling. Weight losses due to drying were recorded and the toxin contents were calculated based on the original material subjected to study. All samples were milled to a particle size of ~1 mm and stored at −18 °C before analysis.

### 5.2. Chemicals and Reagents

Mycotoxin standards (NIV, SCIRP, DON, T2 TETRAOL, FUS-X, MAS, 15-ADON, 3-ADON, T2 TRIOL, NEO, DAS, and HT-2, T-2) were purchased from Sigma (Deisenhofen, Germany). Acetonitrile, hexane, ethanol, methanol, ethyl acetate, anhydrous sodium sulphate, and other chemicals were obtained from Merck (Darmstadt, Germany) and were analytical grade. Trifluoracetic anhydride (TFAA), used as a derivatization reagent, was purchased from Pierce (Rockford, IL, USA). 

### 5.3. Extraction, Clean-up and Analysis

Extraction and clean-up procedures were carried out as described previously in detail by Schollenberger et al. [37]. Briefly, an amount of 10 g of each sample was extracted in a 250 mL screw-capped bottle using a mixture of acetonitrile–water (75:25 *v*/*v*) and a rotary shaker followed by filtration with a folded filter. From the filtrate, an aliquot of 63 mL was subjected for 10 s to a liquid–liquid extraction using hexane (50 mL) and a separating funnel followed by the addition of 60 mL of ethanol and the evaporation to dryness of the aqueous phase with a rotary evaporator with 40 °C in a water bath. The residue was further dissolved in 2.5 mL of methanol, ultrasonicated, and centrifuged for 10 min at 4000 rpm. A clean-up step was performed through a solid-phase extraction by the mean of a florisil and a cation-exchange cartridge. All samples were subjected to gas chromatography-mass spectrometry (GC-MS) analysis. Derivatization was carried out with TFFA. The GC-MS analysis was performed with a Magnum–Ion-Trap-System (Finnigan, Bremen, Germany) operated in the chemical ionization mode (CI) using isobutane as reactant gas and the temperature of the ion trap was 190 °C. A DB-5 MS phase (30 m × 0.25 mm and 0.25 µm film thickness) was the capillary column used and the carrier gas was helium. The temperature for the injection port was 260 °C and the injection volume was 1 µL. The injection mode was splitless, with valve closed for 20 s, afterwards it is split 60 mL/min. Column temperature was set initially at 90 °C for 2 min, then increased to 275 °C at 23°/min, maintained for 2 min at 275 °C, increased to 290 °C at 30°/min, and kept 15 min at 290 °C. The transfer line temperature was adjusted to 270 °C. The maximum ionization time was 1500 µs and 80 ms the maximum reaction time. The ionization level was 25 u and the reaction level 40 u, the 250 u was the reagent ion eject level, the reagent ion eject adjust at 100% with 9000 µs for the reagent reaction time. 

Validation of the method followed the criteria of the EU Commission Decision, 2002/657/EC [55] for linearity, accuracy, and repeatability. Matrix-assisted calibration curve was performed for each studied mycotoxin and provided good linearity when compared with the standard calibration curve, therefore, the matrix effect was assessed for each mycotoxin at the same concentration levels. The matrix-assisted calibration curve was prepared by adding a standard solution of trichothecenes at six concentration levels, from 10 to 2000 µg/kg using blank purified wheat extract. The regression coefficients (r2) of all calibration curves were higher than 0.994, demonstrating good linearity for the proposed method. Detection limits were assessed at a signal-to-noise ratio of 3:1 in samples spiked at the lowest validated level, using purified wheat extract previously analyzed and free of contamination with studied mycotoxins. Detection limits for DON, NIV, SCIRP, T-2 TETRAOL, FUS-X, MAS, 15-ADON, 3-ADON, T-2 TRIOL, NEO, DAS, HT-2, and T-2 were at 10, 20, 5, 45, 10, 2, 4, 9, 6, 4, 6, 2, and 3 µg/kg, quantification limits were 33, 62, 18, 137, 32, 6, 13, 29, 20, 14, 21, 6, and 11 µg/kg and were at a signal to noise ratio of 6:1. The levels of LOQ are lower than the maximum residue limits established by EU for these mycotoxins [24]. Mass ranges used for quantification can be found in Table 9. 

Accuracy was verified by measuring the recoveries from spiked blank samples of wheat, bread, and pasta at levels of 500 µg/kg and 200 µg/kg and their preparation followed the protocol applied for samples. Precision of the method was expressed as relative standard deviation (% RSD) and it was assessed by repeatability, evaluated in four determinations in a single day (n = 4). Results for both groups of toxins investigated were not corrected for recovery.

### 5.4. Statistics

Regarding the occurrence of trichothecenes in bakery products, pasta, and wheat samples, statistical tests were applied to differentiate the levels of these mycotoxins. Collected data were subjected to statistical analysis using IBM SPSS Statistics 20 software. The normality of the groups was verified using the Shapiro–Wilk test. For the descriptive statistics, the mean values, and the standard deviations for the normally distributed data, as well as medians and percentiles (25–50–75) for data that were not normally distributed, were determined. One-way ANOVA (analysis of variance) with post hoc Tukey HSD (honestly significant difference) test were used to make multiple comparisons. The Kruskal–Wallis test was used for the data that was not normally distributed. We used The Bonferroni test for pairwise comparisons. 

## Figures and Tables

**Figure 1 toxins-15-00466-f001:**
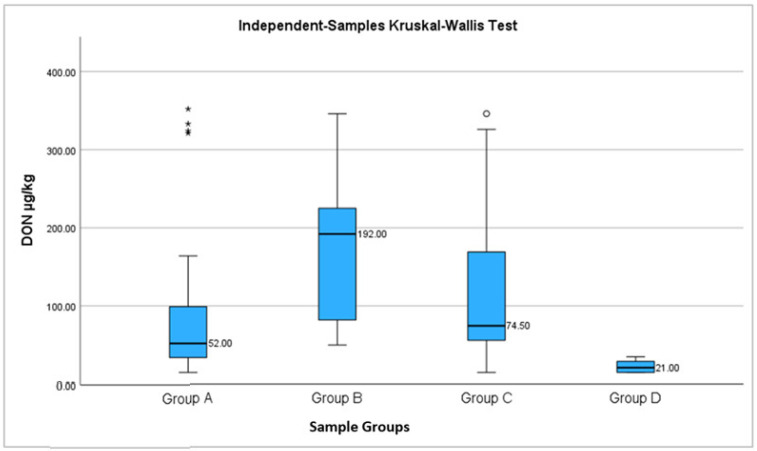
The Boxplot compares the medians between group A (white bread), B (half brown bread), C (brown bread), and D (pasta), regarding DON concentration [µg/kg]; ◦ - represents outlier (value that do not falls in the whisker bar); * - represents extreme outlier (value more than three times the height of the boxes).

**Figure 2 toxins-15-00466-f002:**
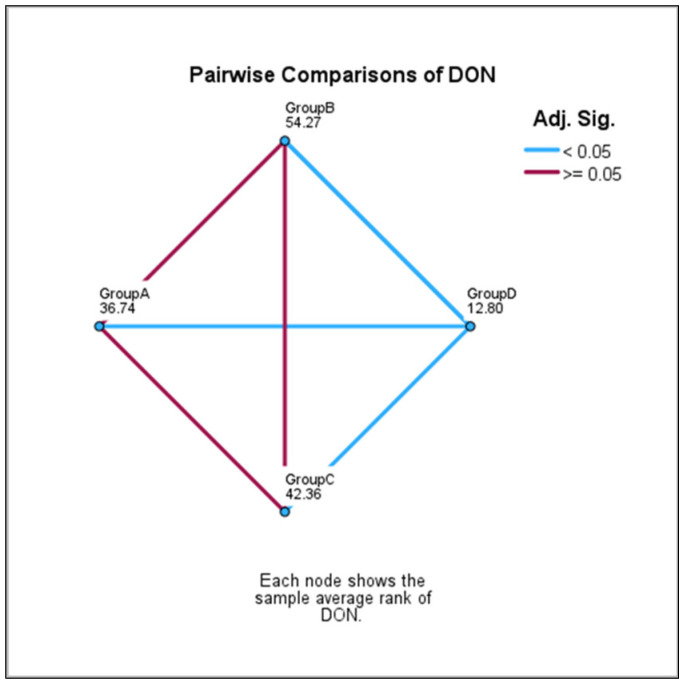
Comparisons of DON between each group; for *p* < 0.05 significant differences; for *p* ≥ 0.05 non-significant differences.

**Figure 3 toxins-15-00466-f003:**
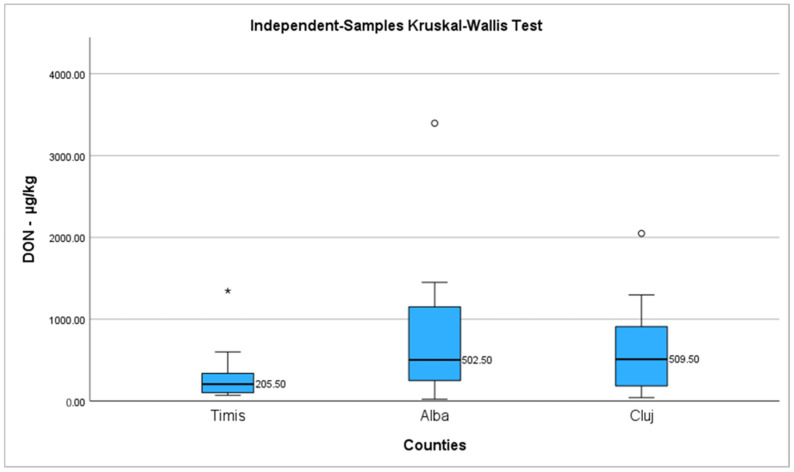
The Boxplot compares the median values of the groups for DON concentration [µg/kg] in wheat samples coming from Timis, Alba, and Cluj counties; ◦ - represents outlier (value that do not falls in the whisker bar); * - represents extreme outlier (value more than three times the height of the box).

**Figure 4 toxins-15-00466-f004:**
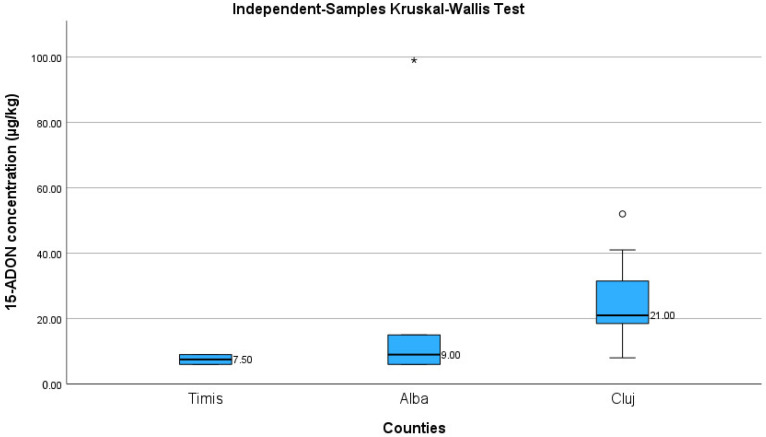
The Boxplot compares the median values between 15-ADON concentration [µg/kg] in wheat samples coming from Timis, Alba, and Cluj counties; ◦ - represents outlier (value that do not falls in the whisker bar); * - represents extreme outlier (value more than three times the height of the box).

**Table 1 toxins-15-00466-t001:** Recoveries and standard deviations (n = 4) of 13 trichothecenes in different matrices spiked at levels of 500 µg/kg and 200 µg/kg.

Toxin	Detection Limit (µg/kg)	Spiking Concentration (µg/kg)	Recoveries (%) and Standard Deviations (%)		
			Wheat	Bread	Pasta
NIV	20	500	55 ± 7.9	68 ± 14.7	57 ± 6.5
		200	64 ± 5.3	68 ± 4.5	69 ± 6.2
FUS-X	10	500	69 ± 10.6	88 ± 4.6	68 ± 7.3
		200	71 ± 3.9	91 ± 6.1	85 ± 6.9
DON	10	500	95 ± 3.4	88 ± 2.6	103 ± 4.9
		200	88 ± 7.9	93 ± 7.1	94 ± 8.2
15-ADON	4	500	83 ± 5.2	97 ± 4.4	81 ± 0.5
		200	90 ± 8.1	93 ± 5.4	91 ± 3.3
3-ADON	9	500	63 ± 10.1	86 ± 7.1	64 ± 3.4
		200	69 ± 9.2	88 ± 11.3	77 ± 11.9
T-2	3	500	83 ± 10.6	103 ± 8.2	79 ± 8.6
		200	81 ± 6.2	94 ± 4.1	91 ± 5.8
HT-2	2	500	75 ± 4.9	87 ± 3.8	77 ± 7.1
		200	72 ± 8.2	83 ± 4.5	79 ± 6.3
T-2 TRIOL	6	500	65 ± 9.6	79 ± 6.1	67 ± 9.1
		200	68 ± 8.5	77 ± 10.1	67 ± 4.4
T-2 TETRAOL	45	500	55 ± 12.1	64 ± 7.3	55 ± 13.4
		200	60 ± 9.2	62 ± 8.7	58 ± 10.2
SCIRP	5	500	63 ± 0.6	78 ± 10.3	62 ± 9.8
		200	62 ± 4.3	67 ± 6.9	62 ± 11.2
MAS	2	500	70 ± 4.7	89 ± 5.1	70 ± 5.5
		200	67 ± 5.5	86 ± 4.5	71 ± 4.8
DAS	6	500	76 ± 15.1	107 ± 6.5	70 ± 5.3
		200	76 ± 10.2	104 ± 8.9	72 ± 6.8
NEO	4	500	80 ± 11.8	99 ± 6.7	78 ± 9.1
		200	78 ± 9.4	102 ± 6.1	76 ± 5.7

**Table 2 toxins-15-00466-t002:** Summary of the occurrence of trichothecenes type A and Type B in analyzed samples.

Sample Type	Toxin Detected	No. of Positive Samples	Range (µg/kg)
Group A	DON	29	15–352
	HT-2	1	3
	T-2	1	5
Group B	DON	13	50–346
Group C	DON	14	15–326
Group D	DON	15	15–35
Group E			
Subgroup E_1_	DON	12	70–1346
	15-DON	2	6–9
	HT-2	7	3–7
Subgroup E_2_	DON	8	21–3395
	15-DON	5	6–99
	DAS	1	19
	HT-2	4	5–8
	T-2	1	7
Subgroup E_3_	DON	18	41–2048
	15-DON	8	8–52
	HT-2	10	3–18
	NIV	1	30

**Table 3 toxins-15-00466-t003:** Descriptive statistics regarding DON [µg/kg] in bakery products and pasta.

		Group A	Group B	Group C	Group D
N		29	13	14	15
Mean		96.65	178.38	114.92	22.13
Median		52.00	192.00	74.50	21.00
Std. Deviation		103.01	102.16	105.13	7.27
Range		337.00	296.00	331.00	20.00
Minimum		15.00	50.00	15.00	15.00
Maximum		352.00	346.00	346.00	35.00
Percentiles	25	33.00	78.00	51.75	15.00
	50	52.00	192.00	74.50	21.00
	75	101.50	268.00	173.75	30.00

Group A—white bread; Group B—half brown bread; Group C—brown bread; Group D—pasta; N—number of positive samples.

**Table 4 toxins-15-00466-t004:** Pairwise comparisons between groups regarding DON concentrations [µg/kg].

Sample1-Sample2	Test Statistic	Std. Error	Std. Test Statistic	Sig.	Adj. Sig.^a^
GroupD-GroupA	23.94	6.55	3.65	<0.001	0.002 **
GroupD-GroupC	29.55	7.66	3.85	<0.001	0.001 ***
GroupD-GroupB	41.46	7.81	5.30	<0.001	0.000 ***
GroupA-GroupC	−5.61	6.71	−0.83	0.403	1.000 NS
GroupA-GroupB	−17.52	6.88	−2.54	0.011	0.065 NS
GroupC-GroupB	11.91	7.94	1.50	0.134	0.802 NS

a—significance values have been adjusted by the Bonferroni correction for multiple tests, NS—non significant, * *p* < 0.05, ** *p* < 0.01, *** *p* < 0.001.

**Table 5 toxins-15-00466-t005:** Descriptive statistics regarding DON [µg/kg] in wheat samples coming from three different counties.

		Timis	Alba	Cluj
N		12	8	18
Mean		315.16	903.00	604.11
Median		205.50	502.50	509.50
Std. Deviation		359.96	1100.04	521.50
Range		1276.00	3374.00	2007.00
Minimum		70.00	21.00	41.00
Maximum		1346.00	3395.00	2048.00
Percentiles	25	100.75	220.75	169.25
	50	205.50	502.50	509.50
	75	387.00	1300.50	933.75

N—number of samples.

**Table 6 toxins-15-00466-t006:** Descriptive statistics regarding 15-ADON [µg/kg] in wheat samples coming from three different counties.

		Timis	Alba	Cluj
N		2	5	8
Mean		7.50	27.00	25.25
Median		7.50	9.00	21.00
Std. Deviation		2.12	40.41	14.15
Range		3.00	93.00	44.00
Minimum		6.00	6.00	8.00
Maximum		9.00	99.00	52.00
Percentiles	25	6.00	6.00	17.75
	50	7.50	9.00	21.00
	75	.	57.00	36.25

N—number of samples.

**Table 7 toxins-15-00466-t007:** Descriptive statistics regarding HT-2 [µg/kg] in wheat samples coming from three different counties.

		Timis	Alba	Cluj
N		7	4	10
Mean		5.00	6.25	9.20
Median		5.00	6.00	9.50
Std. Deviation		1.29	1.50	4.31
Range		4.00	3.00	15.00
Minimum		3.00	5.00	3.00
Maximum		7.00	8.00	18.00
Percentiles	25	4.00	5.00	5.75
	50	5.00	6.00	9.50
	75	6.00	7.75	11.25

N—number of samples.

**Table 8 toxins-15-00466-t008:** Multiple comparisons based on advanced statistics with Post Hoc Test Tukey HSD.

(I) Counties	(J) Counties	Mean Difference (I-J)	Std. Error	Sig.	95% Confidence Interval	
					Lower Bound	Upper Bound
Timis	Alba	−1.25	2.00	0.809 NS	−6.36	3.86
	Cluj	−4.20	1.57	0.040 *	−8.22	−0.17
Alba	Timis	1.25	2.00	0.809 NS	−3.86	6.36
	Cluj	−2.95	1.89	0.289 NS	−7.78	1.88
Cluj	Timis	4.20	1.57	0.040 *	0.17	8.22
	Alba	2.95	1.89	0.289 NS	−1.88	7.78

NS—non significant; * the mean difference is significant at the 0.05 level.

**Table 9 toxins-15-00466-t009:** Mass ranges used for quantification of the investigated trichothecenes.

Mycotoxin	Mass Ranges Used for Quantification
DON	471–473
15-ADON	471–473
3-ADON	531–533
NIV	583–585
FUS-X	529–531
T-2	401–403
HT-2	455–457
T-2 TRIOL	569–571
T-2 TETRAOL	569–571
DAS	403–405
MAS	457–459
SCIRP	411–413
NEO	401–403

## Data Availability

No new data were created or analyzed in this study. Data sharing is not applicable to this article.

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
