# Peer review of "Occurrence of Types A and B Trichothecenes in Cereal Products Sold in Romanian Markets"

_toxins, 2023, doi:10.3390/toxins15070466_

Round 1
Reviewer 1 Report
The paper entitled: Study on occurrence of type A and type B trichothecenes in wheat, bakery products and pasta from Romanian market, deals with the study of detection and evaluation of thirteen trichothecenes in wheat, bread and bakery products and pasta by means of gas chromatography-MS. The authors report that from the total of 121 samples subjected to analysis, 90.08% were contaminated with trichothecenes, DON represented the predominant mycotoxin, occurring in 90.62% of bakery products which contain white flour, 92.85% of bakery products with whole wheat flour, 100% of bakery products with rye flour, 78.94% of pasta samples and in 90.47% of wheat simples, concluding that samples represented by bakery products containing whole wheat flour were contaminated in higher concentrations than samples from white flour, but none of them was exceeding maximum level imposed by regulation and that evaluation of multi-mycotoxins in different food matrices offers a reliable tool for risk assessment. The paper is correct in the development, only using standard procedures for extraction and detection of the mycotoxins in a relatively high number of simples. Some statistical methods are used to compare appearances of the toxins in the samples. The paragraph from lines 128 to 137 is repeated from the paragraph from lines 116 to 125, therefore it should be deleted to avoid repetition. Some recommendations about the sampling, frequency and control of samples from markets should be added as a way to prevent human exposure, specially in cases where toxins are found over the legally permitted amounts. With this corrections the paper is publishable.
The paper entitled: Study on occurrence of type A and type B trichothecenes in wheat, bakery products and pasta from Romanian market, deals with the study of detection and evaluation of thirteen trichothecenes in wheat, bread and bakery products and pasta by means of gas chromatography-MS. The authors report that from the total of 121 samples subjected to analysis, 90.08% were contaminated with trichothecenes, DON represented the predominant mycotoxin, occurring in 90.62% of bakery products which contain white flour, 92.85% of bakery products with whole wheat flour, 100% of bakery products with rye flour, 78.94% of pasta samples and in 90.47% of wheat simples, concluding that samples represented by bakery products containing whole wheat flour were contaminated in higher concentrations than samples from white flour, but none of them was exceeding maximum level imposed by regulation and that evaluation of multi-mycotoxins in different food matrices offers a reliable tool for risk assessment. The paper is correct in the development, only using standard procedures for extraction and detection of the mycotoxins in a relatively high number of simples. Some statistical methods are used to compare appearances of the toxins in the samples. The paragraph from lines 128 to 137 is repeated from the paragraph from lines 116 to 125, therefore it should be deleted to avoid repetition. Some recommendations about the sampling, frequency and control of samples from markets should be added as a way to prevent human exposure, specially in cases where toxins are found over the legally permitted amounts. With this corrections the paper is publishable.
Reviewer 2 Report
Mycotoxins in agricultural commodities, foods, and feeds have long been considered of being a higher risk to human and animal health. It is of great significance for the determination simultaneously of multi-mycotoxins in wheat, bakery products, and pasta using GC-MS. The results offer a reliable tool for daily monitoring, and risk assessment of mycotoxins in different food matrices. However, there were some questions in the manuscript that the authors should explain before this work can be accepted.
Line 428, “Extraction, clean-up and analysis”, the analysis method as a determination for mycotoxins should offer some information: column, chromatographic condition, etc..
The reference mass spectrum parameters of each mycotoxin molecule should be provided.
Reviewer 3 Report
The manuscript is appropriate for publication to the journal after the author do major revisions. The following are the things that authors should consider.:
1. It is sporadically included into the text but not displayed in the manuscript in the same way how the performance characteristics were assessed. It is necessary to present the studies for recoveries, repeatability, and limit of detection in more detail in the Materials and Methods section.
2. The authors should consider if the limit of detection estimation through the pure substance it is appropriate due to the matrix effect.
3. Since May 2023, the European Regulation 1881/2006 is no longer in effect. European Regulation 915/2023 revokes it. In order for the work to be uploaded to the new regulation, authors should revise the text using the new regulation.
4. Please revise all the manuscript for appropriate significant figures. In the manuscript, it is inconsistent. Except for LOD, when two significant figures are appropriate, three significant figures are acceptable in the majority of cases.
Reviewer 4 Report
This article investigated the occurrence of 13 trichothecenes in 5 group samples collected from Romanian market. However, the whole manuscript is not clear for me, the major problem is the novelty and significance of this work since the detection methods have already been reported in a lot of previous studies, the discussion of the result is superficial data statement and the references are out of date. Some questions need to be clearly answered, listed as follows:
1. Please explain the novelty and significance of this work in the Introduction.
2. Please explain why D3G was not detected, which is not consistent with the introduction of the article.
3. Please supplement the data of the method validation, such as the linearity of the detection method, the recoveries of different concentrations in various matrices, and so on.
4. Please add a summary table of the occurrence of mycotoxins in different samples from Romanian market.
5. The recoveries of NIV varied from 55%-68% on different matrices is not very good, please clarify how to calculate the NIV concentrations in positive samples.
6. In the results and discussion section, DON concentration has been repeatedly discussed. Please simplify and modify this part of discussion.
7. Lines 105-106: from 64% (T-2 TETRAOL) to 107% (DAS) with standard deviations from 2.6% (DON) to 14.7% (NIV), the data was not consistent with the Table 1 in wheat.
8. Lines 116-125 should be deleted.
9. Please define the abbreviations of word, such as SCIRP, MAS et. al.
10. Some references are out of date and should be revised.
No
Reviewer 5 Report
The paper reports a study on the occurrence of type A and type B trichothecenes in wheat, bakery products and pasta from Romanian market. Specifically, thirteen trichothecenes pertaining to type A and type B in 121 samples, divided in 5 groups were assessed. As far as the analytical approach is concerned, GC-MS technique was applied for the mycotoxin determination in the investigated samples. Surprisingly, the results indicated contamination in 90.08% of the analyzed samples.
The work is surely of interest since mycotoxins in agricultural commodities are considered of being a risk to human and animal health. However, there are some issues to be considered prior to eventual publication.
The most critical point is related to the section. 5.3. Extraction, clean-up and analysis. It looks like that the authors did not adhere to any international guidelines and some validation parameters have been seriously omitted, e.g. linearity, accuracy, matrix affects, etc. Recovery at a concentration of 500 µg/kg is insufficient.
5.2. Chemicals and Reagents. The selection of the mycotoxin standards reported should be clearly emphasized in the section 3. Discussion
English is fine although is it can be further improved.
Abstract: Potential application of the present study on other samples needs to be emphasized.
Moderate revision.
Round 2
Reviewer 3 Report
I have no other comment. In my opinion the manuscript is ready for publication.
Author Response
Thank you very much for your professional evaluation, for time and effort, as well as for your positive feedback!
The final version of the manuscript comprises some minor changes which were requested by the Academic editor. The manuscript file is attached below.

Reviewer 4 Report
Sorry, but I still can't accept the novelty of the present work since there are too many reports on this aspect.
It is OK.
Author Response
Thank you very much for your professional evaluation, as well as for time and effort!
The final version of the manuscript comprises some minor changes which were requested by the Academic editor. The manuscript file is attached below.

Reviewer 5 Report
Despite the lines reported in the rebuttal are wrong and the text was not highlighted, I managed to check the revised version of the manuscript. Most of the criticisms raised have been solved.
Minor issues detected.
Author Response
Thank you very much for your professional evaluation, for time and effort, as well for your positive feedback! The authors would like to apologize for the messy tracks.
The final version of the manuscript comprises some minor changes which were requested by the Academic editor. The manuscript file is attached below.
